# Encapsulation of D-Limonene into O/W Nanoemulsions for Enhanced Stability

**DOI:** 10.3390/polym15020471

**Published:** 2023-01-16

**Authors:** Md Sohanur Rahman Sohan, Samar Elshamy, Grace Lara-Valderrama, Teetach Changwatchai, Kubra Khadizatul, Isao Kobayashi, Mitsutoshi Nakajima, Marcos A. Neves

**Affiliations:** 1Graduate School of Science and Technology, University of Tsukuba, 1-1-1 Tennodai, Tsukuba, Ibaraki 305-8577, Japan; 2School of Integrative and Global Majors, Tsukuba Life Science Innovation Program, University of Tsukuba, 1-1-1 Tennodai, Tsukuba, Ibaraki 305-8577, Japan; 3Food Research Institute, NARO, 2-1-12 Kannondai, Tsukuba, Ibaraki 305-8642, Japan; 4Graduate School of Life and Environmental Science, University of Tsukuba, 1-1-1 Tennodai, Tsukuba, Ibaraki 305-8577, Japan; 5Alliance for Research on the Mediterranean and North Africa (ARENA), University of Tsukuba, 1-1-1 Tennodai, Tsukuba, Ibaraki 305-8577, Japan; 6Faculty of Life and Environmental Sciences, University of Tsukuba, 1-1-1 Tennodai, Tsukuba, Ibaraki 305-8577, Japan

**Keywords:** D-limonene, O/W emulsion, triglycerides, environmental stress, storage stability, retention

## Abstract

The present study aimed to investigate the physical stability in terms of (droplet size, pH, and ionic strength) and chemical stability in terms of (retention) of D-limonene (LM) in the nanoemulsions after emulsification as well as after storing them for 30 days under different temperatures (5 °C, 25 °C, and 50 °C). LM is a cyclic monoterpene and a major component extracted from citrus fruits. The modification of disperse phase with soybean oil (SB) and a nonionic emulsifier (Tween 80) was adequate to prepare stable LM-loaded nanoemulsions. LM blended with SB-loaded nanoemulsions were stable against droplet growth over pH (3–9) and ionic strength (0–500 mM NaCl). Regarding long-term storage, the prepared nanoemulsions demonstrated excellent physical stability with droplet size ranging from 120–130 nm during 30 days of storage at both 5 °C and 25 °C; however, oiling off started in the emulsions, which were stored at 50 °C from day 10. On the other hand, the retention of LM in the emulsions was significantly impacted by storage temperature. Nanoemulsions stored at 5 °C had the highest retention of 91%, while nanoemulsions stored at 25 °C had the lowest retention of 82%.

## 1. Introduction

D-limonene (LM) is a functional monoterpene and a primary flavor compound found in the epicarp of citrus fruits such as lemon, lime, orange, grapefruit, etc. Among two enantiomers, (R)- and (S)- limonene, found in nature, D-limonene was used in this study, which is an (R)- enantiomer. LM has plenty of advantages, including its use as a flavor and aroma enhancer in various foods, beverages, perfumes, soaps, and other products. According to some previous studies, LM also has a wide range of health benefits. For instance, it has antioxidant properties letting it resist the impact of free radicals that harm DNA and induce cancer as well as LM has numerous chemo-preventive properties which can inhibit the growth of different sorts of cancer, such as gastric cancer, prostate cancer, liver cancer, and lung cancer [1,2,3]. It has also been found that LM has antimicrobial properties that can work against foodborne pathogens [4]. Besides, LM contains anti-inflammatory properties and has been counted in bronchial asthma treatment [5]. In addition, LM is safe to consume because it is listed as generally considered safe (GRAS) [6].

However, some significant challenges when applying LM as a bioactive compound to a food complex include its low water solubility (13.8 mg/L at 25 °C) [7], hydrophobic nature [8], and easy degradation by oxidation under room temperature (25 °C) [9] as well as low viscosity (0.79 mPa.s). Many studies have been conducted to overcome those difficulties and to find the appropriate nano-delivery systems for LM, such as nanoemulsions (phase inversion method, organogel-based, and whey protein-maltodextrin assisted), nanoliposomes (thin film dehydration method), and solid lipid nanoparticles (hot, high-pressure homogenization utilizing glycerol monostearate) [7]. 

Nano-encapsulation is one of the promising technology which may solubilize essential oils in the aqueous phase by creating small oil droplets dispersed in the aqueous phase that contains surfactants. Previous studies also reported the advantages of applying nano-encapsulation technology to encapsulate essential oils such as cumin, coriander [10], clove, cinnamon, and lavender [11]. Nevertheless, emulsions are a combination of two immiscible liquids whereby the help of an emulsifier, one is dispersed into another. Primarily emulsions can be classified into oil-in-water (O/W) emulsions and water-in-oil (W/O) emulsions. LM’s lipid solubility led them to consider oil-in-water (O/W) nanoemulsions where LM dispersed in the oil phase & coated with an emulsifier. Nanoemulsions have the capacity to deliver a higher concentration of the disperse phase while using less emulsifier, as well as easy manufacture, improved bioactive compounds, controlled droplet size, and stability [12,13,14,15]. Even though there are many advantages of nanoemulsions, Ostwald ripening is one of the destabilizing mechanisms that can still impact the stability of nanoemulsions. Hence, it is possible to impede the Ostwald ripening by choosing the proper emulsifier and modifying oil composition by adding long-chain triglyceride (LCT). Soybean oil (SB) was used to modify the disperse phase in this study since it contains many long-chain triglycerides, for instance, linoleic acid, linolenic acid, and oleic acid. The emulsifier is crucial in preventing the recoalescence of newly formed droplets, and modifying oil can assist in stopping other destabilizing mechanisms.

Most previous research explained the formulation characteristics of the LM-loaded nanoemulsions prepared using different techniques and their physical stability over time. However, to our knowledge, there is limited or nonexistent published research on the retention of LM after the formulation of nanoemulsions stabilized by soybean oil alongside a non-ionic emulsifier (Tween 80) and following their long-term storage at different temperatures. Therefore, the objective of this research was to investigate the effect of environmental stress, including pH, ionic strength, and temperatures (5, 25, and 50 °C) on the stability of the LM-loaded nanoemulsions and to evaluate the retention of LM in the nanoemulsions following emulsification as well as after storing them for 30 days under different temperatures. The findings from this study could be used for future applications, such as extending the shelf-life of perishable agricultural products.

## 2. Materials and Methods

### 2.1. Materials

D-limonene (LM), soybean oil (SB), caprylic acid (CP) and polyoxyethylene (80) sorbitan monooleate (Tween 80), sodium hydroxide, hydrochloric acid, and sodium azide were purchased from Fujifilm Wako Pure Chemical Corporation (Osaka, Japan). The current study prepared all the solutions and emulsions with water purified by an Ultrapure Water System (Arium^®^ Comfort II, Sartorius Co., Göttingen, Germany). In addition, all the chemicals used in this study were analytically graded and applied accordingly throughout experimentation.

### 2.2. Viscosity Measurements

The effect of LM-SB weight ratio (0:10, 2.5:7.5, 5:5, 7.5:2:5, 9:1, and 10:0) in disperse phase, and emulsions viscosity was measured by Vibro Viscometer (SV-10, A & D Co. Ltd., Tokyo, Japan). Average values from at least three sample estimations were used for analysis.

### 2.3. Interfacial Tension Measurements

The interfacial tension between the disperse phase containing LM pure or blended with SB and the aqueous phase comprised of ultra-pure water with or without 1% (*w*/*w*) Tween 80 were determined by applying the pendant drop method (DM-501, Kyowa Interface Science Co., Ltd. Saitama, Japan) at room temperature (25 ± 2 °C). Before interfacial tension measurements, densities of the dispersed and aqueous phases were measured by a density/specific gravity meter (DA-130N, Kyoto Electronics Co., Ltd., Kyoto, Japan). For interfacial tension measurements, the aqueous phase was placed in a syringe and pushed out from a pendant drop at the tip of a stainless-steel needle inside a glass cell surrounded by the dispersed phase, LM pure or blended with SB. During these measurements, image analysis was done by interface measurement & analysis software (FAMAS, Kyowa Interface Science Co., Ltd. Saitama, Japan). This analysis software calculates the interfacial tension of the samples by the Young-Laplace equation. Each sample was estimated at least three times (10 measurements each), and analysis was performed using the average values.

### 2.4. Emulsions Preparation

O/W emulsions were prepared by homogenizing the 10% (*w*/*w*) disperse phase (LM, SB, or CP) with a 90% (*w*/*w*) aqueous phase. The disperse phase consisted of either LM pure or blended with CP at weight ratios of 5:5 and with SB at different weight ratios, as follows: 2.5:7.5, 5:5, 7.5:2.5, and 9:1, respectively. Besides, the aqueous phase was comprised of ultrapure water containing 1.0% (*w*/*w*) polyoxyethylene (20) sorbitan monooleate (Tween 80) as the emulsifier, and 0.02% (*w*/*w*) sodium azide to inhibit microbial proliferation during storage. A coarse emulsion premix was prepared using a high-speed homogenizer (Polytron, PT3100, Kinematica-AG, Luzern, Switzerland) at 7000 rpm for 5 min. Further homogenization was done by passing the coarse emulsion through a high-pressure homogenizer (NanoVater, NV 200, Yoshida Kikai, Nagoya, Japan) at 100 MPa for three passes.

### 2.5. Emulsion Stability Testing

#### 2.5.1. Effect of pH

The measurements of the effect of pH on the emulsions were done by adjusting the pH of the prepared fresh emulsions to a specified level (pH 3–9) using Sodium hydroxide (1 M) and/or Hydrochloric acid (1 M) solution. Then, the samples were shifted into glass-made test tubes and kept for 24 h at room temperature (25 ± 2 °C) for analysis.

#### 2.5.2. Effect of Ionic Strength

The effect of ionic strength on the freshly prepared emulsions was done by adding different concentrations of NaCl stock solution to achieve 0–500 mM. The samples were put into glass test tubes and stored for 24 h at room temperature (25 ± 2 °C) before analysis.

#### 2.5.3. Long-Term Storage

The prepared emulsions were placed into glass test tubes for 10 days at 25 °C to determine the ideal disperse phase modifier. The emulsions with optimal disperse phase modifiers were stored at different temperatures (5, 25, or 50 °C) for 30 days to evaluate their long-term stability.

### 2.6. Droplet Size Analysis

The droplet size of the formulated nanoemulsions was evaluated using a laser diffraction particle size analyzer (LS 13320, Beckman Coulter Ltd., Brea, CA, USA). The refractive indices of LM 1.471, SB 1.432, and CP 1.428 were measured using a pocket refractometer (PAL-RI, ATAGO Co. Ltd., Tokyo, Japan). Droplet size estimations were stated as Sauter mean diameter (*d*_3,2_) as indicated in Equation (1), where *n_i_* is the droplet diameter (*d_i_*) [16]. Average values from at least three sample estimations were used for analysis.
(1)$$d_{3,2}=\frac{\mathrm{Volume}}{\mathrm{Surface}\;\mathrm{area}}=\frac{\sum_{i=1}n_{i}d_{i}^{3}}{\sum_{i=1}n_{i}d_{i}^{2}}$$

### 2.7. Gas Chromatography (GC) Analysis of LM Retention in the Emulsions over Time

The prepared emulsions were put in firmly sealed glass test tubes covered with aluminum foil and stored at 5 °C (inside a temperature-controlled refrigerator) and 25 °C (inside a dark temperature-controlled incubator) for 30 days. First, the test tube containing LM and SB-loaded nanoemulsions was lightly shaken for 10 s to homogenize the sample. Following that, a 100 µL sample was added into a 10 mL glass tube comprising 4 mL of methanol with 0.5 g of (NH_4_)_2_SO_4_. Prior to filtering with a 0.45 µm syringe filter, the glass tube containing the solution was vortexed for 1 min. In order to perform the GC analysis, 1.5 mL of the sample solutions were finally transferred to the injection vials. The samples were analyzed by a gas chromatograph (Shimadzu GC-2025) fitted with a capillary column (Zebron^TM^ZB-WAXplus^TM^, 30.0 m × 0.25 mm × 0.25 µm) attached to an AOC-20i automatic liquid injector (10 µL syringe) and flame ionization detector (FID). The temperature program of the GC was initially set to 60 °C for 2 min to stabilize the column temperature; then, the column was heated with a heating rate of 30 °C min^−1^ until the temperature reached 210 °C and held for 3 min. Finally, the column was cooled down with a cooling rate of 30 °C min^−1^ until the temperature returned to 60 °C and held for 1 min. The injector and ion source temperatures were set at 220 °C and 270 °C, respectively, with the 10:1 split rate and helium as the carrier (30 mL min^−1^). LM was determined through a calibration curve created using standard LM. The limonene (analytical standard) (CAS number: 5989-27-5) was purchased from Sigma-Aldrich Co., Ltd. (St. Louis, MO, USA). The i-PeakFinder algorithm was used to do automatic peak integration. The sample preparation procedure was adapted from [17]. LM retention was calculated utilizing the following equation:(2)$$\mathrm{Limonene}\;\mathrm{retention}\;(\%)=\frac{L_{t}}{L_{0}}\times 100$$
where, *L*_0_ and *L_t_* are the concentrations of LM in emulsions after emulsification and each interval time (5 days), respectively.

### 2.8. Statistical Analysis

Each experiment was performed in triplicate, and the data are reported as average ± standard deviation. One-way analysis of variance (ANOVA) was done using SPSS Statistics (IBM Statistics 28, Armonk, NY, USA) to characterize the viscosity, interfacial tension, and effect of disperse phase on LM-loaded emulsions stability. Duncan’s multiple ranges were used to compare the means at a 95% confidence level. The different letters indicate significant differences (*p* < 0.05).

## 3. Results and discussion

### 3.1. Effect of Disperse Phase on LM-Loaded Emulsions Stability

The physical stability of O/W emulsions formulated using LM pure or blended with CP or SB at weight ratios of 5:5 containing 1.0% (*w*/*w*) Tween 80 under room temperature (25 °C) was investigated. The changes in the droplet sizes of the prepared emulsions are illustrated in Figure 1a. The average droplet size of the freshly prepared emulsions loaded with LM pure or blended with CP or SB was 482.16 nm, 433 nm, and 120.66 nm, respectively. Although the droplet size of the freshly prepared emulsions was small on day 0, the droplet size of the emulsions, other than LM blended with SB, started to increase from the next day. Notably, the droplet size of only LM-loaded emulsions significantly increases (*p* < 0.05) from the following day, possibly due to Ostwald ripening. As shown in Figure 1b, emulsions loaded with LM blended SB had smaller droplets with a single peak around 147 nm, indicating the homogeneity of the emulsions. Bimodal distributions were shown in both LM pure, and LM blended with CP-loaded emulsions. Only LM-loaded emulsions showed the first peak at around 111 nm and the second at around 544 nm. LM blended with CP-loaded emulsions showed the first peak at around 122 nm and a second one at around 1300 nm, which portrayed the heterogeneity of the emulsions (Figure 1b). The droplet size distribution of the only LM-loaded emulsion also suggests that the Laplace pressure’s difference between larger and smaller droplets contributes to the Ostwald ripening occurrences [18]. Ostwald ripening and coalescence are possible destabilizing mechanisms of the growth in oil droplet size. Furthermore, it is evidenced that Ostwald ripening is a common phenomenon that is observed in essential oils, short-chain triglycerides, and flavor oil-in-water emulsions [19,20]. It occurs when oil molecules diffuse from small to large droplets across the aqueous phase, causing the larger droplets to consume the smaller ones [21]. It is demonstrated that adding highly hydrophobic molecules, such as triglycerides, can control the Ostwald ripening of emulsions prepared with oils that are not highly water soluble [22]. Regarding determining the influence of triglyceride, and fatty acid on the stability of the emulsions, SB (as it contains long-chain triglyceride (LCT)), and CP (as a medium-chain fatty acid (MCFA)), respectively, added in the disperse phase to prepare the emulsions. Both emulsions in which CP and SB are blended with LM showed stability upon initial storage. Nevertheless, the droplet size of CP-assisted LM emulsions slightly increased from day 2. Additionally, phase separation was observed in the emulsions where CP was blended with LM after 10 days of storage (Figure 1c). The less viscosity of the disperse phase (LM:CP- 5:5; 1.25 mPa.s) allowed for easier diffusion and coalescence-induced instability [23]. In contrast, the emulsions prepared from SB blended with LM showed stability during the entire period of storage (Figure 1a). The presence of various LCTs in SB, including linoleic acid, linolenic acid, and oleic acid, might assist in stabilizing formed nanoemulsions due to their high hydrophobicity and high viscosity (47.35 mPa.s) [24].

Therefore, SB continued to be used in modifying disperse phase for further experiments since SB blended with LM emulsions showed stability within the time range tested.

### 3.2. Effect of SB Concentration on Disperse Phase and Emulsions Viscosity

Gradually increasing SB concentration significantly increased the viscosity (*p* < 0.05) of disperse phase since the viscosity of SB is high (47.35 mPa.s). However, despite utilizing varied concentrations in the disperse phase, it was found that after measuring the viscosity of freshly prepared emulsions, the viscosity was relatively low (1–2 mPa.s), which may be due to the use of a small amount of disperse phase (10% (*w*/*w*)) in the emulsions. In addition, it is reported that the emulsions’ viscosity will increase when the droplets start to coalesce [25]. In this case, emulsions’ viscosity was measured immediately after emulsification, and therefore it showed stable viscosity. Furthermore, since the viscosity of disperse phase (LM:SB- 10:0; 0.79 mPa.s) was close to the continuous phase (1 mPa.s), and LM has a slight solubility in water (13.8 mg/L); thus, LM readily diffuses in the water. Based on the Stokes-Einstein equation, it is found that diffusivity is inversely proportional to viscosity. Hence, the reduced viscosity of the disperse phase enhances the diffusivity of LM [23]. The instability of only LM-loaded and CP-assisted LM emulsions in Section 3.1 can also be explained by the phenomena of the lower viscosity of disperse phase. 

### 3.3. Effect of Tween 80 and Disperse Phase Modification on Interfacial Tension

Interfacial tension is crucial in preparing an emulsion because a more physically stable emulsion can be obtained when its value is relatively low. Therefore, the effect of the LM-SB weight ratio and Tween 80 on interfacial tension between the dispersed and continuous phases was investigated. The standard interfacial tension between disperse phase consisting of limonene and soybean oil mixture and ultrapure water without emulsifier was 24.2 mN/m. Though when 1% (*w*/*w*) of Tween 80 was added to the continuous phase, the interfacial tension reduced to 8 mN/m. The weight ratio of LM and SB in disperse phase did not significantly change the interfacial tension (*p* < 0.05). Whereas the addition of Tween 80 affected the interfacial tension, which may be due to its higher water miscibility and fast adsorbing to the lipid droplets’ surface [26].

### 3.4. Effect of Environmental Stresses on LM-Loaded Emulsions Stability

#### 3.4.1. Effect of pH

The stability of the emulsions under different pH levels has to be investigated, considering commercial application in the food and beverages industries. The effect of pH on the stability of the emulsions can be assessed by estimating the mean droplet size of the emulsions adjusted in the pH range of 3–9 after 24 h of storage at 25 °C. This evaluation was conducted only after storing for 24 h because pH was considered a variable parameter in this case but time was not, and 24 h is required to allow for any reactions between the emulsions and pH ions [27,28]. As shown in Figure 2a, the emulsions prepared using different oil concentrations showed stability over the whole pH range (3–9) without indicating droplet growth or noticeable instability. Most previous studies showed that the pH had minimal influence on the stability of the emulsions that incorporated non-ionic emulsifiers [27,28,29,30]. The droplet aggregation was principally controlled by steric repulsion because of the comparably large hydrophilic (polyoxyethylene) head groups of the adsorbed Tween 80 molecules [31]. Since the emulsions in this study were prepared using a non-ionic surfactant, Tween 80, there was no effect of pH on the physical stability of emulsions.

#### 3.4.2. Effect of Ionic Strength

The emulsion stability against ionic strength has to be studied regarding varying amounts of minerals that may be introduced to emulsion-based products. The effect of ionic strength on the stability of the emulsion was investigated by adding various quantities of salt (0–500 mM NaCl) to all the emulsions samples after 24 h of storage at 25 °C. This estimation was performed after storing for 24 h, considering NaCl as a variable parameter but not time. Besides, 24 h is needed to allow for any reactions between the emulsions and NaCl ions [27,28]. No visible changes were found in droplet size throughout the whole range of salt levels examined (Figure 2b). The non-ionic nature of Tween 80 forms steric repulsion between oil droplets that keep the droplet stable even after increasing the NaCl concentrations [16,29]. Therefore, the effect of NaCl on emulsions prepared using Tween 80 loaded with different concentrations of LM and SB was negligible.

#### 3.4.3. Effect of Storage Time and Temperature

The droplet sizes of the emulsions formulated using 1% (*w*/*w*) of Tween 80 loaded with different concentrations of LM and SB under different temperatures (5 °C, 25 °C, and 50 °C) were investigated. No change in droplet size and visual appearance were observed in the emulsions over 30 days of storage, both at 5 °C (Figure 3a,b) and 25 °C (Figure 4a,b), respectively. The average droplet size of freshly prepared emulsions loaded with different concentrations of LM and SB in disperse phase, A: LM 2.5% + SB 7.5%, B: LM 5% + SB 5%, C: LM 7.5% + SB 2.5%, and D: LM 9% + SB 1%, was 124 nm, 122.66 nm, 124.33 nm, and 125.66 nm, respectively. The droplet size remained similar over the entire storage period, at 5 °C and 25 °C, regardless of the LM and SB concentration in disperse phase. Overall, LM blended with SB-loaded emulsions were stable because SB has bunches of long-chain triglycerides (LCTs), for instance, linoleic acid, linolenic acid, and oleic acid, which have high viscosity as well as high hydrophobicity that might assist in stabilizing formed nanoemulsions [24]. On the other hand, it was found to increase in droplet size in the emulsion samples stored at 50 °C (Figure 5a), and oiling off started from day 10 (Figure 5b). Higher temperatures reduce oil’s viscosity, leading to more droplet collisions and widening the density difference between dispersed and continuous phases. In addition, non-ionic emulsifiers’ relative solubility in water decreases as temperature rises, which accelerates droplet coalescence and facilitates demulsification [32]. The emulsions were not physically stable at 50 °C; hence, further chemical stability investigation was not done at this storage temperature.

### 3.5. Chemical Stability of LM-Loaded Emulsions over Time under Different Temperatures

The chemical stability of LM-loaded emulsions was evaluated based on the emulsification efficiency to retain LM till the end of the experimental storage period (30 days) at different temperatures (5 and 25 °C).

For that purpose, LM concentration was measured before and after the emulsification process to determine the initial concentration in our experiment. It was found that about 70–80% of LM remained after the emulsification process in all the samples (A, B, C, and D), indicating some reduction during emulsion preparation. The emulsion with the lowest LM concentration (sample A) was noticed to have the highest reduction in its LM value. In contrast, the emulsion with the highest LM concentration (sample D) showed the lowest reduction in the LM value. Essential oils are known for their sensitivity to the high shear and pressure that were used during the emulsification process, which could be the main reason for LM loss in this step (Table 1) [33].

The thermal stabilities of LM-loaded emulsions at different temperatures (5 and 25 °C) are presented in Figure 6a,b, respectively. By tracking the emulsions LM concentration over 30 days of storage, it was found that samples (A, B, C, and D) kept at 25 °C were able to retain (88.87, 78.16, 81.04, and 84.02%) of LM, respectively, at the end of storage period. In contrast, samples kept at 5 °C showed outstanding retention (95.21, 84.97, 93.98, and 92.49%), respectively, which is efficiently higher than the samples kept at 25 °C. LM is considered a volatile compound due to its low molecular weight (136.23 g/mol) and high vapor pressure at room temperature (25 °C). Besides, Hassanzadeh et al. (2022) reported that a higher temperature caused a greater release of the volatile compound from the emulsions [34]. In addition, increasing the temperature has an impact on accelerating LM release from the emulsions. It could be explained due to increasing the motional energy, which increases the molecule collision and accelerates the LM release. Furthermore, LM is a lipophilic compound with a log *p* value of 4.57, indicating its hydrophobicity and affinity for lipids. In addition, LM has a lower density (0.84 g/mL) than water (0.99 g/mL), preventing it from mixing with water and causing it to flow to the outer phase. Because of this, LM may have volatilized while being stored at 25 °C, causing them to retain less LM than emulsions that were stored at 5 °C. Besides, since the emulsions were kept in a closed system, LM as a gas phase in the headspace would be equilibrium; thus, oxidation may occur. Subsequently, the oxidation rate also has a dependency on the temperature [35]. Therefore, the increasing storage temperature causes LM to move out from the disperse phase and undergo greater oxidation. Although both emulsions stored at 5 °C and 25 °C had similar droplet sizes (Figure 3a and Figure 4a), however, it was found that LM losses were higher at 25 °C than at 5 °C, probably due to the volatilization as well as oxidation of the compound [17,35,36]. The results revealed outstanding chemical stability of the LM-loaded emulsions indicating a suitable emulsification method for LM. Furthermore, the results suggest that it is recommended to store the LM emulsions at 5 °C to achieve the maximum chemical stability and highest retention concentration.

## 4. Conclusions

This study investigated the physical and chemical stability of LM-loaded nanoemulsions. We found that the addition of SB in disperse phase alongside Tween 80 helped to produce stable LM-loaded nanoemulsions, whereas CP had no effect. Emulsions prepared with LM blended SB were stable across the whole range of pH (3–9), and NaCl (0–500 mM) levels tested. In addition, the emulsions stored at 5 °C and 25 °C were remarkably physically stable in terms of the droplet size range (122–130 nm) over the entire period of 30 days of storage but highly unstable at 50 °C. However, the retention of LM in the nanoemulsions was greatly influenced by the storage temperature. The nanoemulsions stored at 5 °C demonstrated excellent retention of 91% in contrast to 82% at 25 °C. Based on our results, LM-loaded nanoemulsions at 5 °C could maintain maximum chemical stability. Overall, our findings provide new information regarding the ideal disperse phase modifier to prepare stable LM-loaded nanoemulsions and the optimum temperature to store these emulsions for maximum LM retention for possible future applications in the food and beverage industries.

## Figures and Tables

**Figure 1 polymers-15-00471-f001:**
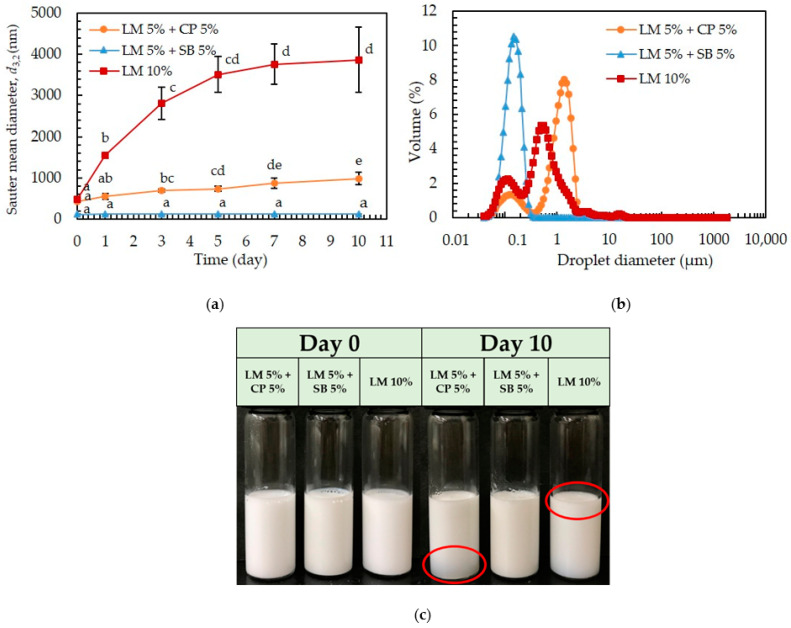
Effect of the dispersed phase in the D-limonene-loaded emulsions. (**a**) Storage stability at 25 °C (**b**) Droplet size distribution of freshly prepared emulsions, and (**c**) The photos of the emulsions on days 0 and 10. The different letters indicate significant differences (*p* < 0.05) by Duncan’s test.

**Figure 2 polymers-15-00471-f002:**
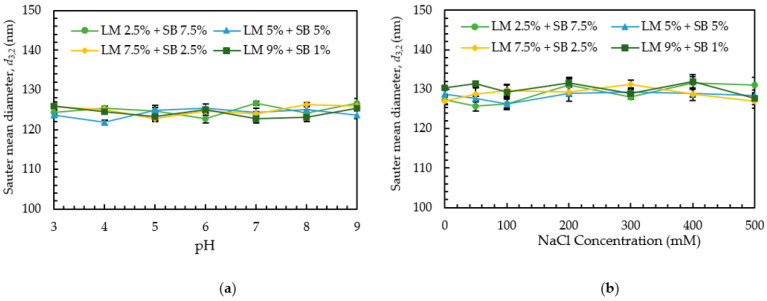
(**a**) Effect of pH and (**b**) NaCl on the Sauter mean diameter (*d*_3,2_).

**Figure 3 polymers-15-00471-f003:**
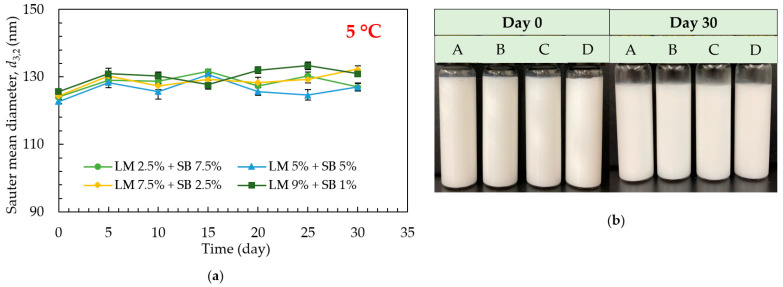
Effect of storage time, temperature, and SB concentration on the physical stability of the emulsions loaded with LM during 30 days of storage at 5 °C. (**a**) Sauter mean diameter (*d*_3,2_) and (**b**) The photos of the emulsions on days 0 and 30 where A: LM 2.5% + SB 7.5%, B: LM 5% + SB 5%, C: LM 7.5% + SB 2.5%, and D: LM 9% + SB 1%.

**Figure 4 polymers-15-00471-f004:**
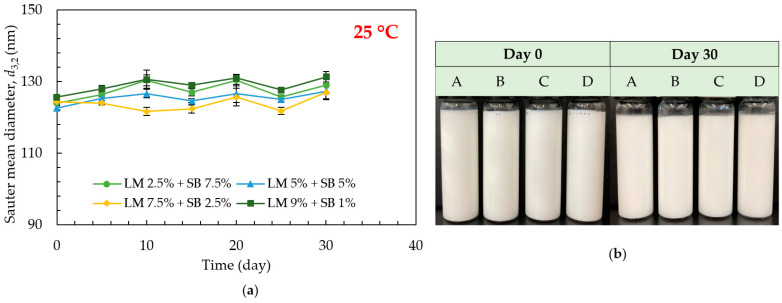
Effect of storage time, temperature, and SB concentration on the physical stability of the emulsions loaded with LM during 30 days of storage at 25 °C. (**a**) Sauter mean diameter (*d*_3,2_) and (**b**) The photos of the emulsions on days 0 and 30 where A: LM 2.5% + SB 7.5%, B: LM 5% + SB 5%, C: LM 7.5% + SB 2.5%, and D: LM 9% + SB 1%.

**Figure 5 polymers-15-00471-f005:**
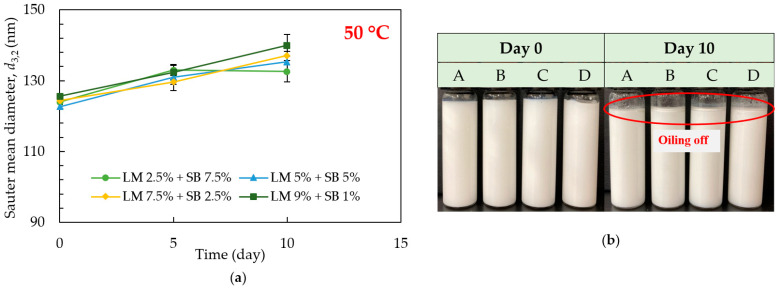
Effect of storage time, temperature, and SB concentration on the physical stability of the emulsions loaded with LM during 10 days of storage at 50 °C. (**a**) Sauter mean diameter (*d*_3,2_) and (**b**) The photos of the emulsions on days 0 and 10 where A: LM 2.5% + SB 7.5%, B: LM 5% + SB 5%, C: LM 7.5% + SB 2.5%, and D: LM 9% + SB 1%.

**Figure 6 polymers-15-00471-f006:**
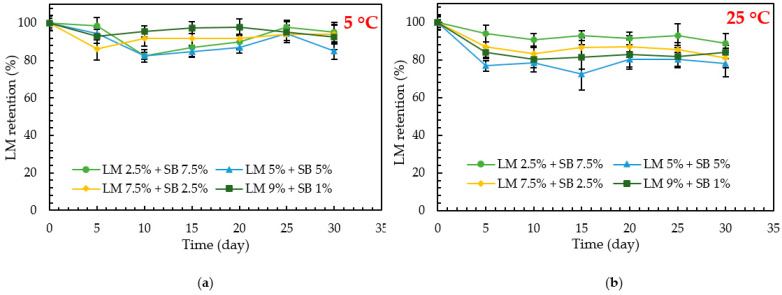
Retention of D-limonene in the emulsions during storage at (**a**) 5 °C and (**b**) 25 °C. Data are presented as mean ± standard deviation, *n* = 3.

**Table 1 polymers-15-00471-t001:** Remaining of D-limonene in the emulsions after emulsification.

LM-SB Weight Ratio in Disperse Phase	Sample Code	Initial (g/L)	After Emulsification	
			Day 0, *L_0_* (g/L)	Remaining (%)
2.5:7.5	A	21.25 ± 0.2	14.82 ± 1.2	69.74 ± 0.7
5:5	B	42.50 ± 0.1	34.07 ± 1.1	80.16 ± 0.1
7.5:2.5	C	63.75 ± 0.1	50.68 ± 2.7	79.50 ± 0.8
9:1	D	76.50 ± 0.2	64.21 ± 3.7	83.93 ± 0.7

## Data Availability

Not applicable.

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
