# Peer review of "Encapsulation of D-Limonene into O/W Nanoemulsions for Enhanced Stability"

_polymers, 2023, doi:10.3390/polym15020471_

Round 1

Reviewer 1 Report

Dear Authors, 

The Manuscript entitled "Encapsulation of D-limonene into O/W Nanoemulsions for Enhanced Stability" presents some seroius weekness.

- In the tile you report "... Nanoemulsion..." however, your O/W emulsions present micrometric size;

- The Abstract needs to be completely re-writed. Ex: "... for a certain period of time..." is not scientitifc. Please add real values. 

- Why did you measure the viscosity? Whis is the relevance of this analysis in this Manuscript?

- Where is the statistical analysis with relative discussion? 

- Interfacial tension. Where is the standars with water? 

- In geberal, very low quality discussion of the results compared to the litterature. 

The Manuscript is not suitable for a publication. 

Author Response

First of all, the authors would like to express sincere thanks to the reviewer for kindly spending your valuable time on a thorough review of our manuscript. We believe that your comments and suggestion would lead to the improvement of our paper. Please kindly see below, for a point-by-point response to your comments and questions, where the page and line numbers mentioned hereafter refer to the number in the revised version of the manuscript.

The Manuscript entitled "Encapsulation of D-limonene into O/W Nanoemulsions for Enhanced Stability" presents some serious weakness.

Response: Thank you for your valuable comments in order to improve the understanding of our work. Your comments and suggestion would guide the improvement of our manuscript.

In the title you report "... Nanoemulsion..." however, your O/W emulsions present micrometric size

Response: Thank you for raising this important issue. However, during the preliminary experiments to determine the ideal disperse phase modifier for the stability of LM-loaded nanoemulsions, the droplet size was found to be micrometric in Figure 1. Since we aimed to enhance the stability of LM-loaded emulsions, thus that condition was not considered for further investigation as they were not stable within the time range tested.

In addition, some other characteristics, such as the high-energy method preparation, which begins with a high-speed homogenizer followed by a high-pressure homogenizer, and thermodynamic instability, can also define the nanoemulsions.

Furthermore, the emulsions' droplet sizes, which were utilized to assess their long-term stability were in the 120–130 nm range, which is also within the range of nanoemulsions. Therefore, overall, we reported our emulsion as "nanoemulsion" as well as kept it in the title. We hope the reviewer will kindly understand our reasoning for stating nanoemulsion in the title and throughout the manuscript.

The Abstract needs to be completely re-writed. Ex: "... for a certain period of time..." is not scientific. Please add real values.

Response: Thank you for your insightful comment. The real value 30 days was written in the abstract. In addition, the whole abstract was revised following your kind suggestion.

Why did you measure the viscosity? Which is the relevance of this analysis in this Manuscript?

Response: We would like to thank you for asking this important question. Viscosity is a crucial physical parameter regarding formulations and the stability of the emulsions. We presented the viscosity of disperse phase and its effect on the viscosity of the emulsions. Because we found that when the viscosity of disperse phase (0.79 mPa.s) was similar to the continuous phase (1 mPa.a), the emulsions were not stable in the case of 10% (w/w) LM-loaded emulsions in section 3.1.

Generally, if the oil phase's viscosity is close to the continuous phase, the oil will readily diffuse in the water. Since the viscosity of disperse phase (LM:SB- 10:0) (0.79 mPa.s) was close to the continuous phase (1 mPa.s), and LM has a slight solubility in water (13.8 mg/L); therefore, LM also diffuses in the water which results in the instability of 10% (w/w) LM-loaded emulsions in our findings.

Therefore, we measured the viscosity and included Figure 2 (revised) in the manuscript to show that a minor decrease in the disperse phase viscosity has an effect on the stability of the emulsions. We sincerely hope the reviewer will appreciate the justification for both the viscosity study and the decision to keep the figure in the manuscript.

Where is the statistical analysis with relative discussion? 

Response: The authors appreciate the reviewer’s question. Statistical analysis was conducted to assess significant differences among variables of viscosity, interfacial tension, and the effect of disperse phase on LM-loaded emulsions stability. Whereas statistical analysis was not applied to the long-term emulsion's stability and LM-retention studies. In addition, following your kind suggestion, statistical analysis with relative discussion was added where it was required throughout the manuscript.

Interfacial tension. Where is the standard with water? 

Response: Thank you for your valuable comment. The standard interfacial tension between disperse phase and ultra-pure water without emulsifier is 24.24 ± 0.2 mN/m which is already added on page 7, lines 246-247.

In general, very low quality discussion of the results compared to the literature. 

Response: The authors acknowledge the respected reviewer's opinion. More clarification was added throughout the manuscript for further improvement.

Reviewer 2 Report

The manuscript "Encapsulation of D-limonene into O/W Nanoemulsions for Enhanced Stability" presents a study of the physical and chemical stability of LM-loaded nanoemulsions function of different environmental conditions.

The study is well conducted, the investigations are well described, and the outcome is properly discussed.

However, my suggestion is that the authors have to better point the necessity of this kind of study.

Also, there are some minor flows in english (ex. at page 7), which must be corrected. 

Author Response

First and foremost, the authors would like to express their sincere gratitude to the reviewer for taking the time to review the manuscript carefully and providing important comments, which have improved the overall quality of our manuscript. We really appreciate your kind encouragement. Please kindly see below, for a point-by-point response to your comments and questions, where the page and line numbers mentioned hereafter refer to the number in the revised version of the manuscript.

The manuscript "Encapsulation of D-limonene into O/W Nanoemulsions for Enhanced Stability" presents a study of the physical and chemical stability of LM-loaded nanoemulsions function of different environmental conditions.

The study is well conducted, the investigations are well described, and the outcome is properly discussed.

Response: The authors deeply appreciate your kindness in spending your valuable time reviewing this manuscript. We are so grateful for your encouraging feedback.

However, my suggestion is that the authors have to better point the necessity of this kind of study.

Response: The authors acknowledge the reviewer’s opinion. There is limited or nonexistent published research on the retention of LM after emulsification assisted by soybean oil alongside a non-ionic emulsifier (Tween 80) and following their long-term storage at different temperatures. In addition, the authors also believe that the findings from this study could be used for future applications such as extending the shelf-life of perishable agricultural products (page 2, lines 78-88).

Also, there are some minor flows in english (ex. at page 7), which must be corrected. 

Response: We are grateful to the reviewer for raising this important point. They have been corrected following your kind comment.

Reviewer 3 Report

The manuscript “Encapsulation of D-limonene into O/W Nanoemulsions for Enhanced Stability” focusses in the encapsulation of D-limonene (LM) in oil-in-water (O/W) emulsions to enhance their physical and chemical stability by tuning the oil phase. Even though the authors clearly showcase the objectives and present the details of the experimental work, I believe some further clarifications have to be made to publish the manuscript. Moreover, I would suggest language polishing throughout the manuscripts. Clarifications/comments:

Lines 84-85:

If the authors would like to give the viscosity values of the different emulsion components, they should rewrite the sentence.

Lines 89-92:

Is it required to mention the features of the viscosimeter?

Line 112:

Polyoxyethylene (20) or (80)????

Line 113:

Why was considered the addition of sodium azide? There were no proteins in the systems? Where they expecting microbial growth?

Lines 120-123:

It is not clear if the emulsions were prepared first and the pH was adjusted after or the emulsions were prepared with a continuous phase at the proper pH. It seems to me the first scenario, but it would be good to clarify. 

Lines 125-127:

The same comment as for the pH: how was the ionic strength adjusted? Was NaCl added after emulsion preparation? 

Lines 129-130:

The authors have used different storage times, it would be convenient that they list (1 day, 10 days, 30 days) all the conditions in the materials and methods section. 

Line 158:

Why irradiation time?

Lines 172-173:

The authors mention that emulsions had considerable heterogeneity or good homogeneity. They should explain more. Are they referring to the dispersity of the droplet size? Moreover, when they provide the droplet size of the emulsions, two of them have a size of about 0,5 um. Therefore, I think it would be advisable to mention that even though the value are in the range of 500 nanometers, the size is not what is expected for a nanoemulsion.

Line 183:

It should be Figure 1b

Lines 196-201:

I think the authors fail to note that one of the emulsions (LM 10%) goes through a destabilization process for 10 days. In fact, according to Figure 1 (a) the Sauter diameter is 3 times bigger after 1 day and it multiplies by 10 at the end of the stability study. I think it is worthy to comment on this, since they want to enhance the stability by missing with other oils. 

Figure 1:

The caption should be corrected, (a) and (b) are swapped.

Lines 215-216:

I think the assumption is wrong. Since the oil fraction is 10% (w/w) the viscosity of the emulsion will be very similar to the viscosity of the continuous phase. From Figure 2 it can be observed that the viscosity of LM is quite similar to water. Apart from that, the figure does not add any relevant information. 

Lines 225-228:

The authors should explain why the concentration of SB can modify the interfacial tension. Depending on the composition of the oil phase there will be a different interfacial tension. However, the parameter that changes the interfacial tension is the addition of a surfactant as well as its concentration.  I would suggest not to include Figure 3, since it does not add any relevant information.

Lines 242-253:

For the effect of the environmental stresses, the authors start with the effect of the pH. Why did they decide to follow the samples only during 1 day (the same applies to ionic strength). During the initial tests to define the  best conditions for stability, they followed the emulsions for 10 days. What is the reason to reduce the time span? Why the authors did not include a control sample with LM 10%?? It seems that after 1 days all the samples perform equally well, therefore if the 10% LM gives a higher load, why not use it??

Lines 265-284:

As for the effect of T of storage, why the authors decided to reduce the time span to 10 days for 50ºC? The increase in the droplet size does not seem highly significant. It goes from about 120 nanometers to 140 nanometers (tops). For the studies at 5 and 25 ºC the droplet size was kept under 130 nanometers, but it is really a short difference. Why the study at 50ºC was not continued further?

Moreover, why a control sample with 10% LM was not included in the trials?

Table 1:

The table dies not include any errors

Lines 320-321:

The assumption about osmotic pressure is totally wrong. The osmotic unbalance is quite important in double emulsions (like the ones reported in reference 29), but for single emulsions is not a key issue.

Lines 314-334:

Regarding the chemical stability measured in this work I have two main concerns. My first concern is related with how the samples of the stability study were stored. I can assume that the samples stored at 5ºC were in the fridge, and therefore protected from the light most of the time. How about the samples at 25 ºC, were they protected from the light? D-limonene can go through oxidation caused by light.

My second concern is about how the authors calculate the “retention rate”, which is not defined anywhere in the paper. Which is the value that they use as initial concentration? The initial load or the amount remaining after emulsification? They should carefully define the parameter (giving an equation always helps to clarify) and then the values given will have more sense. When they mention the retention rate, it should be also considered the amount that is lost during emulsification. The values of a “retention rate” at more than 90% are not that unusual at low temperatures, where the emulsion stability is higher. 

Based on all these comments, I think the authors need to rewrite the conclusions. They mention that is highly recommended to store LM-loaded nanoemulsions in the refrigerator, but that was expected. 

Author Response

First, the authors would like to express their deepest gratitude to the reviewer for taking the time to thoroughly review and appraise your detailed comments and suggestions to improve our manuscript's quality. Please kindly see below, for a point-by-point response to your comments and questions, where the page and line numbers mentioned hereafter refer to the number in the revised version of the manuscript.

The manuscript “Encapsulation of D-limonene into O/W Nanoemulsions for Enhanced Stability” focusses in the encapsulation of D-limonene (LM) in oil-in-water (O/W) emulsions to enhance their physical and chemical stability by tuning the oil phase. Even though the authors clearly showcase the objectives and present the details of the experimental work, I believe some further clarifications have to be made to publish the manuscript. Moreover, I would suggest language polishing throughout the manuscripts. Clarifications/comments:

Response: We would like to express our thanks to the reviewer for kindly spending your valuable time on a thorough review of our manuscript. Following your kind comment, a clarification has been made throughout the manuscript. We sincerely believe that your comments and suggestion would lead to the improvement of our paper.

Lines 84-85: If the authors would like to give the viscosity values of the different emulsion components, they should rewrite the sentence.

Response: We acknowledge the reviewer's insightful comment. The viscosity values were already deleted.

Lines 89-92: Is it required to mention the features of the viscosimeter?

Response: We are grateful to the reviewer for raising this point. We do agree with your opinion. Therefore, the features of the viscometer have already been deleted.

Line 112: Polyoxyethylene (20) or (80)????

Response: Thank you for the insightful comment. Polyoxyethylene (20) sorbitan monooleate was used in this research. Another name for polyoxyethylene (20) sorbitan monooleate is Tween 80. On the other hand, Polyoxyethylene (20) sorbitan monolaurate is known as Tween 20.

Line 113: Why was considered the addition of sodium azide? There were no proteins in the systems? Where they expecting microbial growth?

Response: We would like to thank you in advance for this important question. We do agree with the insightful comment. Although the system doesn't contain any protein, storing them at different environmental conditions, such as 25 or 50 °C for a long time, makes them more susceptible to microorganisms. In addition, the emulsions contain soybean oil and emulsifier, which may be a potential nutrient for microorganisms. Therefore, sodium azide was added to the emulsions for long-term storage stability evaluation.

Lines 120-123: It is not clear if the emulsions were prepared first and the pH was adjusted after or the emulsions were prepared with a continuous phase at the proper pH. It seems to me the first scenario, but it would be good to clarify. 

Response: Thank you very much for raising these important points. We would like to apologize for this confusion. We do agree with your comments. The emulsions were prepared first and then adjusted to different pH levels to observe the effect of pH on the freshly prepared emulsions. It has already been clarified on page 3, lines 131-133.

Lines 125-127: The same comment as for the pH: how was the ionic strength adjusted? Was NaCl added after emulsion preparation? 

Response: We are so grateful to the reviewer for pointing out these issues. The authors would like to apologize for this confusion. Similar to pH, the emulsions were prepared first and then ionic strength was adjusted. It has already been clarified on page 3, lines 136-137.

Lines 129-130: The authors have used different storage times, it would be convenient that they list (1 day, 10 days, 30 days) all the conditions in the materials and methods section. 

Response: We are grateful to the reviewer for the insightful comment. In our preliminary research, we examined the storage stability for 10 days to determine the ideal disperse phase modifier. However, we stored the prepared emulsions with optimal disperse phase modifiers for 30 days to evaluate long-term stability, which has already been clarified on page 3, lines 140-143.

Line 158: Why irradiation time?

Response: We acknowledge the reviewer’s insightful comment and raising this important point. We would like to apologize for this mistake sincerely. Since we measured the limonene retention every five days, therefore it will be each interval time which has already been clarified on page 4, lines 172-173.

Lines 172-173: The authors mention that emulsions had considerable heterogeneity or good homogeneity. They should explain more. Are they referring to the dispersity of the droplet size? Moreover, when they provide the droplet size of the emulsions, two of them have a size of about 0,5 um. Therefore, I think it would be advisable to mention that even though the value are in the range of 500 nanometers, the size is not what is expected for a nanoemulsion.

Response: We acknowledge the reviewer's insightful comment. We would like to apologize for the unclear explanation. We do agree with your opinion that the droplet size distribution can also explain the dispersity of the droplet in the emulsions. A detailed explanation regarding droplet distribution was revised and presented on page 5, lines 190-200.

Line 183: It should be Figure 1b

Response: Thank you very much for pointing out this error. It has already been corrected on page 5, line 198.

Lines 196-201: I think the authors fail to note that one of the emulsions (LM 10%) goes through a destabilization process for 10 days. In fact, according to Figure 1 (a) the Sauter diameter is 3 times bigger after 1 day and it multiplies by 10 at the end of the stability study. I think it is worthy to comment on this, since they want to enhance the stability by missing with other oils. 

Response: The authors would like to appreciate this insightful comment. Based on our preliminary experiments, we found it exceedingly difficult to prepare stable LM-loaded emulsions, as demonstrated in Figure 1a, where the droplet size remarkably increases from the second day. Pure LM-loaded (10% (w/w)) emulsions showed a significant droplet size increase (P < 0.05) by the second day, which may be due to Ostwald ripening, a common destabilization phenomenon seen in flavor oil-in-water emulsions, short-chain triglycerides, and essential oils. Therefore, we decided to modify the disperse to inhibit this destabilization process. Park et al., 2022 also reported that the addition of highly hydrophobic molecules, such as triglycerides, can control the Ostwald ripening of emulsions prepared with oils that are not highly water soluble. Overall, we aimed to enhance the stability of LM-loaded emulsion and soybean oil alongside Tween 80 helped to formulate it.

Reference:

Park SH, Hong CR, Choi SJ. Prevention of Ostwald ripening in orange oil emulsions: Impact of surfactant type and Ostwald ripening inhibitor type. Lwt. 2020;134(September):110180. doi:10.1016/j.lwt.2020.110180.

Figure 1: The caption should be corrected, (a) and (b) are swapped.

Response: We are so grateful for pointing out this mistake. We would like to apologize for this error. The caption has been corrected following your kind comment (page 6, lines 223-224).

Lines 215-216: I think the assumption is wrong. Since the oil fraction is 10% (w/w) the viscosity of the emulsion will be very similar to the viscosity of the continuous phase. From Figure 2 it can be observed that the viscosity of LM is quite similar to water. Apart from that, the figure does not add any relevant information. 

Response: Thank you very much for your insightful comment. We do agree with your valuable comment. We do agree with your valuable comment. Since using a small amount of disperse phase (10% (w/w)), the viscosity of emulsions was similar to the continuous phase. In addition, we would like to keep the revised Figure 2 in the manuscript because it presented the viscosity of disperse phase after modifying the disperse phase and the effect of disperse phase's viscosity in the emulsions. Furthermore, the decreased viscosity of the disperse phase, shown in Figure 2, can also be used to explain the instability of 10% (w/w) LM-loaded emulsions in section 3.1. Since the viscosity of the disperse phase (LM:SB- 10:0) (0.79 mPa.s) was identical to that of the continuous phase (1 mPa.s), thus, limonene diffuses easily in water. In general, the oil will diffuse into the water easily if the viscosity of the oil phase is close to that of the continuous phase. Therefore, we kept Figure 2 (revised) in the manuscript to show that a minor decrease in the disperse phase viscosity has an effect on the stability of the emulsions. We hope the reviewer will kindly understand our reasoning for keeping the figure in the manuscript.

Lines 225-228: The authors should explain why the concentration of SB can modify the interfacial tension. Depending on the composition of the oil phase there will be a different interfacial tension. However, the parameter that changes the interfacial tension is the addition of a surfactant as well as its concentration.  I would suggest not to include Figure 3, since it does not add any relevant information.

Response: We are so grateful to the reviewer for pointing out these issues. We do agree with your valuable comment. Initially we speculate that an increase of SB concentration will decrease interfacial tension, however our result shown that the weight ratio of LM-SB in disperse phase did not significantly change the interfacial tension (P < 0.05). Nevertheless, the addition of Tween 80 affected the interfacial tension. Besides, figure 3 has already been deleted following your kind suggestion.

Lines 242-253: For the effect of the environmental stresses, the authors start with the effect of the pH. Why did they decide to follow the samples only during 1 day (the same applies to ionic strength). During the initial tests to define the  best conditions for stability, they followed the emulsions for 10 days. What is the reason to reduce the time span? Why the authors did not include a control sample with LM 10%?? It seems that after 1 days all the samples perform equally well, therefore if the 10% LM gives a higher load, why not use it??

Response: The authors appreciate the reviewer’s question. We chose to investigate the effect of pH and ionic strength on emulsions stability only after storing them for 1 day because pH or NaCl was considered a variable parameter here but time was not a variable parameter. Besides, 24 h is required to allow for any reactions between the emulsions and ions. In addition, many previous articles also reported that the effect of pH and NaCl on the stability of the emulsions was measured only after 24 h (Taarji et al., 201; Melanie et al., 2020).

As stated in section 3.1, the emulsions containing 10% (w/w) LM were not stable even at neutral pH and no salt; hence, they were not considered for stability evaluation after adjusting pH and ionic strength.

Overall, our goal was to enhance the stability of the emulsions loaded with LM; thus, only the emulsion found stable during initial tests are considered for stability evaluation under environmental stresses such as pH, ionic strength, and temperatures.

References:

Taarji, N.; Rabelo da Silva, C. A.; Khalid, N.; Gadhi, C.; Hafidi, A.; Kobayashi, I.; Neves, M. A.; Isoda, H.; Nakajima, M. Formulation and Stabilization of Oil-in-Water Nanoemulsions Using a Saponins-Rich Extract from Argan Oil Press-Cake. Food Chem. 2018, 246 (November 2017), 457–463. https://doi.org/10.1016/j.foodchem.2017.12.008.

Melanie, H.; Taarji, N.; Zhao, Y.; Khalid, N.; Neves, M. A.; Kobayashi, I.; Tuwo, A.; Nakajima, M. Formulation and Characterisation of O/W Emulsions Stabilised with Modified Seaweed Polysaccharides. Int. J. Food Sci. Technol. 2020, 55 (1), 211–221. https://doi.org/10.1111/ijfs.14264.

Lines 265-284: As for the effect of T of storage, why the authors decided to reduce the time span to 10 days for 50ºC? The increase in the droplet size does not seem highly significant. It goes from about 120 nanometers to 140 nanometers (tops). For the studies at 5 and 25 ºC the droplet size was kept under 130 nanometers, but it is really a short difference. Why the study at 50ºC was not continued further?

Moreover, why a control sample with 10% LM was not included in the trials?

Response: We would like to thank you for raising these questions. Even though the droplet size was relatively small up to 10 days, we nevertheless chose to shorten the period for the emulsions, which were stored at 50 degrees since oiling off was seen in the emulsions. Oiling off is a common destabilization phenomenon in oil-in-water (O/W) emulsions. In addition, it is a standard method for droplet size measurements that the sample must be taken from the middle part of a test tube. Therefore, there was potential that the sample could be taken from the area with smaller droplet sizes despite physical instability.

The sample containing 10% (w/w) LM was not included because it showed instability even after being stored at 25 ºC, which is explained in section 3.1. Therefore, we concluded in section 3.1 that 10% (w/w) LM-loaded emulsions were not considered for further investigation because of their instability.

Table 1: The table does not include any errors

Response: Thank you very much for pointing out this issue. The error data has already been added in Table 1.

Lines 320-321: The assumption about osmotic pressure is totally wrong. The osmotic unbalance is quite important in double emulsions (like the ones reported in reference 29), but for single emulsions is not a key issue.

Response: We do agree with the respected reviewers' comments. The opinion has enlightened us toward a more scientific results explanation. The revised precise explanation has been added on page 11, lines 335-340.

Lines 314-334: Regarding the chemical stability measured in this work I have two main concerns. My first concern is related with how the samples of the stability study were stored. I can assume that the samples stored at 5ºC were in the fridge, and therefore protected from the light most of the time. How about the samples at 25 ºC, were they protected from the light? D-limonene can go through oxidation caused by light.

My second concern is about how the authors calculate the “retention rate”, which is not defined anywhere in the paper. Which is the value that they use as initial concentration? The initial load or the amount remaining after emulsification? They should carefully define the parameter (giving an equation always helps to clarify) and then the values given will have more sense. When they mention the retention rate, it should be also considered the amount that is lost during emulsification. The values of a “retention rate” at more than 90% are not that unusual at low temperatures, where the emulsion stability is higher. 

Response: We do agree with the valuable opinion. Regarding chemical stability evaluation, all the prepared emulsions were put in firmly sealed glass test tubes covered with aluminum foil. Then, they were stored at 5 °C (inside a temperature-controlled refrigerator) and 25 °C (inside a dark temperature-controlled incubator) for 30 days. This information has been added in the materials and methods section on page 4, lines 152-154.

We are so grateful to the reviewer for pointing out this issue. We do agree with your valuable opinion, and we would like to apologize for this typographical error. We did not calculate the "retention rate"; we calculated the "limonene retention" in each interval time (5 days) from equation number 2 (page 4, line 171). In addition, the remaining amount of limonene after emulsification was presented in Table 1, and overall retention during storage for 30 days was calculated from the remaining amount after emulsification.

Based on all these comments, I think the authors need to rewrite the conclusions. They mention that is highly recommended to store LM-loaded nanoemulsions in the refrigerator, but that was expected. 

Response: Thank you very much for your valuable comments throughout the manuscript. We believe that your comments and suggestion would lead to the improvement of our manuscript. The conclusions section has already been revised.

Round 2

Reviewer 1 Report

Dear Authors, 

The Manuscript is now in a better "shape" however, the section "Results adn Discussion" is still low in content. Please, provide a discussion by comparing your data with the litterature. 

Best, 

Author Response

The authors deeply appreciate your kindness in spending your valuable time reviewing this manuscript.

We are so grateful for your encouraging feedback.

In the "Results and discussion" section, additional clarification about the instability of CP-assisted LM emulsions has been added on page 5, lines 214-215. Furthermore, more clarification regarding the effect of soybean oil concentration on disperse phase and emulsions viscosity has been added on page 6, lines 238-241.

Besides, justification about the time duration for the effect of pH and ionic strength on emulsions stability was added on page 7, lines 258-261, and lines 275-278, respectively.

In addition, more explanation with reference regarding chemical stability has been added on page 10, lines 349-356.

Reviewer 3 Report

Lines 215-216: I think the assumption is wrong. Since the oil fraction is 10% (w/w) the viscosity of the emulsion will be very similar to the viscosity of the continuous phase. From Figure 2 it can be observed that the viscosity of LM is quite similar to water. Apart from that, the figure does not add any relevant information. 

Response: Thank you very much for your insightful comment. We do agree with your valuable comment. We do agree with your valuable comment. Since using a small amount of disperse phase (10% (w/w)), the viscosity of emulsions was similar to the continuous phase. In addition, we would like to keep the revised Figure 2 in the manuscript because it presented the viscosity of disperse phase after modifying the disperse phase and the effect of disperse phase's viscosity in the emulsions. Furthermore, the decreased viscosity of the disperse phase, shown in Figure 2, can also be used to explain the instability of 10% (w/w) LM-loaded emulsions in section 3.1. Since the viscosity of the disperse phase (LM:SB- 10:0) (0.79 mPa.s) was identical to that of the continuous phase (1 mPa.s), thus, limonene diffuses easily in water. In general, the oil will diffuse into the water easily if the viscosity of the oil phase is close to that of the continuous phase. Therefore, we kept Figure 2 (revised) in the manuscript to show that a minor decrease in the disperse phase viscosity has an effect on the stability of the emulsions. We hope the reviewer will kindly understand our reasoning for keeping the figure in the manuscript.

I would suggest the authors to include some references on the enhancement of limonene diffusivity with a decrease in the viscosity and comment on that. I still think Figure 2 does not add value to the paper.

Response: The authors appreciate the reviewer’s question. We chose to investigate the effect of pH and ionic strength on emulsions stability only after storing them for 1 day because pH or NaCl was considered a variable parameter here but time was not a variable parameter. Besides, 24 h is required to allow for any reactions between the emulsions and ions. In addition, many previous articles also reported that the effect of pH and NaCl on the stability of the emulsions was measured only after 24 h (Taarji et al., 201; Melanie et al., 2020).

I suggest the authors to include the references in the revised paper adding the comment to support their experimental decision.

As stated in section 3.1, the emulsions containing 10% (w/w) LM were not stable even at neutral pH and no salt; hence, they were not considered for stability evaluation after adjusting pH and ionic strength.

Overall, our goal was to enhance the stability of the emulsions loaded with LM; thus, only the emulsion found stable during initial tests are considered for stability evaluation under environmental stresses such as pH, ionic strength, and temperatures.

References:

Taarji, N.; Rabelo da Silva, C. A.; Khalid, N.; Gadhi, C.; Hafidi, A.; Kobayashi, I.; Neves, M. A.; Isoda, H.; Nakajima, M. Formulation and Stabilization of Oil-in-Water Nanoemulsions Using a Saponins-Rich Extract from Argan Oil Press-Cake. Food Chem. 2018246 (November 2017), 457–463. https://doi.org/10.1016/j.foodchem.2017.12.008.

Melanie, H.; Taarji, N.; Zhao, Y.; Khalid, N.; Neves, M. A.; Kobayashi, I.; Tuwo, A.; Nakajima, M. Formulation and Characterisation of O/W Emulsions Stabilised with Modified Seaweed Polysaccharides. Int. J. Food Sci. Technol. 202055 (1), 211–221. https://doi.org/10.1111/ijfs.14264.

Table 1: The table does not include any errors

Response: Thank you very much for pointing out this issue. The error data has already been added in Table 1.

Not for the initial sample

Lines 320-321: The assumption about osmotic pressure is totally wrong. The osmotic unbalance is quite important in double emulsions (like the ones reported in reference 29), but for single emulsions is not a key issue.

Response: We do agree with the respected reviewers' comments. The opinion has enlightened us toward a more scientific results explanation. The revised precise explanation has been added on page 11, lines 335-340.

I still think it is oxidation of LM that occurs, and this is why the concentration decreases.

Author Response

First and foremost, the authors would like to express their sincere gratitude to the reviewer for taking the time to review and consider your insightful remarks and recommendations in order to improve our manuscript.

Please kindly see below for a point-by-point response to your comments, where the page and line numbers mentioned hereafter refer to the number in the revised manuscript version.

Lines 215-216: I think the assumption is wrong. Since the oil fraction is 10% (w/w) the viscosity of the emulsion will be very similar to the viscosity of the continuous phase. From Figure 2 it can be observed that the viscosity of LM is quite similar to water. Apart from that, the figure does not add any relevant information. 

Response: Thank you very much for your insightful comment. We do agree with your valuable comment. We do agree with your valuable comment. Since using a small amount of disperse phase (10% (w/w)), the viscosity of emulsions was similar to the continuous phase. In addition, we would like to keep the revised Figure 2 in the manuscript because it presented the viscosity of disperse phase after modifying the disperse phase and the effect of disperse phase's viscosity in the emulsions. Furthermore, the decreased viscosity of the disperse phase, shown in Figure 2, can also be used to explain the instability of 10% (w/w) LM-loaded emulsions in section 3.1. Since the viscosity of the disperse phase (LM:SB- 10:0) (0.79 mPa.s) was identical to that of the continuous phase (1 mPa.s), thus, limonene diffuses easily in water. In general, the oil will diffuse into the water easily if the viscosity of the oil phase is close to that of the continuous phase. Therefore, we kept Figure 2 (revised) in the manuscript to show that a minor decrease in the disperse phase viscosity has an effect on the stability of the emulsions. We hope the reviewer will kindly understand our reasoning for keeping the figure in the manuscript.

I would suggest the authors to include some references on the enhancement of limonene diffusivity with a decrease in the viscosity and comment on that. I still think Figure 2 does not add value to the paper.

 Response: The authors appreciate the reviewer’s insightful comment and suggestion. The reference regarding the enhancement of limonene diffusivity with a decrease in the viscosity with further clarification has been added on page 6, lines 238-241. In addition, figure 2 has already been deleted following your kind suggestion.

The authors appreciate the reviewer’s question. We chose to investigate the effect of pH and ionic strength on emulsions stability only after storing them for 1 day because pH or NaCl was considered a variable parameter here but time was not a variable parameter. Besides, 24 h is required to allow for any reactions between the emulsions and ions. In addition, many previous articles also reported that the effect of pH and NaCl on the stability of the emulsions was measured only after 24 h (Taarji et al., 2018; Melanie et al., 2020).

I suggest the authors to include the references in the revised paper adding the comment to support their experimental decision.

 Response: We acknowledge the reviewer's insightful comment and suggestion. Following your kind suggestion regarding supporting experimental decision, the clarification with citation about the effect of pH and ionic strength on emulsions stability was added on page 7, lines 258-261, and lines 275-278, respectively. Finally, the references were also included at the end in the references section.

As stated in section 3.1, the emulsions containing 10% (w/w) LM were not stable even at neutral pH and no salt; hence, they were not considered for stability evaluation after adjusting pH and ionic strength.

Overall, our goal was to enhance the stability of the emulsions loaded with LM; thus, only the emulsion found stable during initial tests are considered for stability evaluation under environmental stresses such as pH, ionic strength, and temperatures.

References:

Taarji, N.; Rabelo da Silva, C. A.; Khalid, N.; Gadhi, C.; Hafidi, A.; Kobayashi, I.; Neves, M. A.; Isoda, H.; Nakajima, M. Formulation and Stabilization of Oil-in-Water Nanoemulsions Using a Saponins-Rich Extract from Argan Oil Press-Cake. Food Chem. 2018246 (November 2017), 457–463. https://doi.org/10.1016/j.foodchem.2017.12.008.

Melanie, H.; Taarji, N.; Zhao, Y.; Khalid, N.; Neves, M. A.; Kobayashi, I.; Tuwo, A.; Nakajima, M. Formulation and Characterisation of O/W Emulsions Stabilised with Modified Seaweed Polysaccharides. Int. J. Food Sci. Technol. 202055 (1), 211–221. https://doi.org/10.1111/ijfs.14264.

Table 1: The table does not include any errors

Response: Thank you very much for pointing out this issue. The error data has already been added in Table 1.

Not for the initial sample

 Response: We are so grateful to the reviewer for pointing out this issue. We would like to apologize for this mistake sincerely. The error data for the initial sample has already been added in Table 1.

Lines 320-321: The assumption about osmotic pressure is totally wrong. The osmotic unbalance is quite important in double emulsions (like the ones reported in reference 29), but for single emulsions is not a key issue.

Response: We do agree with the respected reviewers' comments. The opinion has enlightened us toward a more scientific results explanation. The revised precise explanation has been added on page 11, lines 335-340.

I still think it is oxidation of LM that occurs, and this is why the concentration decreases.

 Response: The authors acknowledge the reviewer's insightful comment. We do agree that oxidation is one of the factors causing the decrease in the LM concentration. The revised clarification about oxidation has been added on page 10, lines 349-356.
